# Evaluating large language models for drafting emergency department encounter summaries

**Christopher Y. K. Williams**[1]*, **Jaskaran Bains**[2], **Tianyu Tang**[2], **Kishan Patel**[2],
**Alexa N. Lucas**[2], **Fiona Chen**[2], **Brenda Y. Miao**[1], **Atul J. Butte**[1], **Aaron E. Kornblith**[1,2]

**1** Bakar Computational Health Sciences Institute, University of California, San Francisco, California, United States of America, **2** Department of Emergency Medicine, University of California, San Francisco, California, United States of America

* cykw2@doctors.org.uk

## Abstract

Large language models (LLMs) possess a range of capabilities which may be applied to the clinical domain, including text summarization. As ambient artificial intelligence scribes and other LLM-based tools begin to be deployed within healthcare settings, rigorous evaluations of the accuracy of these technologies are urgently needed. In this cross-sectional study of 100 randomly sampled adult Emergency Department (ED) visits from 2012 to 2023 at the University of California, San Francisco ED, we sought to investigate the performance of GPT-4 and GPT-3.5-turbo in generating ED encounter summaries and evaluate the prevalence and type of errors for each section of the encounter summary across three evaluation criteria: 1) Inaccuracy of LLM-summarized information; 2) Hallucination of information; 3) Omission of relevant clinical information. In total, 33% of summaries generated by GPT-4 and 10% of those generated by GPT-3.5-turbo were entirely error-free across all evaluated domains. Summaries generated by GPT-4 were mostly accurate, with inaccuracies found in only 10% of cases, however, 42% of the summaries exhibited hallucinations and 47% omitted clinically relevant information. Inaccuracies and hallucinations were most commonly found in the Plan sections of LLM-generated summaries, while clinical omissions were concentrated in text describing patients' Physical Examination findings or History of Presenting Complaint. The potential harmfulness score across errors was low, with a mean score of 0.57 (SD 1.11) out of 7 and only three errors scoring 4 ('Potential for permanent harm') or greater. In summary, we found that LLMs could generate accurate encounter summaries but were liable to hallucination and omission of clinically relevant information. Individual errors on average had a low potential for harm. A comprehensive understanding of the location and type of errors found in LLM-generated clinical text is important to facilitate clinician review of such content and prevent patient harm.

**Data availability statement:** All data supporting the findings described in this manuscript are available in the article, Supplementary Information, or from the corresponding author upon request. The UCSF Information Commons database is available to individuals affiliated with UCSF who can contact the UCSF's Clinical and Translational Science Institute (CTSI) (ctsi@ucsf.edu) or the UCSF's Information Commons team for more information (info. commons@ucsf.edu). If the reader is not affiliated with UCSF, they can contact Atul Butte (atul.butte@ucsf.edu) to discuss official collaboration.

**Funding:** The author(s) received no specific funding for this work.

**Competing interests:** CYKW reports holding equity in Quality Health, Inc. AEK is a co-founder and consultant to CaptureDx. AJB is a co-founder and consultant to Personalis and NuMedii; consultant to Mango Tree Corporation, and in the recent past, Samsung, 10x Genomics, Helix, Pathway Genomics, and Verinata (Illumina); has served on paid advisory panels or boards for Geisinger Health, Regenstrief Institute, Gerson Lehman Group, AlphaSights, Covance, Novartis, Genentech, and Merck, and Roche; is a shareholder in Personalis and NuMedii; is a minor shareholder in Apple, Meta (Facebook), Alphabet (Google), Microsoft, Amazon, Snap, 10x Genomics, Illumina, Regeneron, Sanofi, Pfizer, Royalty Pharma, Moderna, Sutro, Doximity, BioNtech, Invitae, Pacific Biosciences, Editas Medicine, Nuna Health, Assay Depot, and Vet24seven, and several other non-health related companies and mutual funds; and has received honoraria and travel reimbursement for invited talks from Johnson and Johnson, Roche, Genentech, Pfizer, Merck, Lilly, Takeda, Varian, Mars, Siemens, Optum, Abbott, Celgene, AstraZeneca, AbbVie, Westat, and many academic institutions, medical or disease specific foundations and associations, and health systems. AJB receives royalty payments through Stanford University, for several patents and other disclosures licensed to NuMedii and Personalis. AJB's research has been funded by NIH, Peraton (as the prime on an NIH contract), Genentech, Johnson and Johnson, FDA, Robert Wood Johnson Foundation, Leon Lowenstein Foundation, Intervalien Foundation, Priscilla

## Author summary

Large language models (LLMs) possess a range of capabilities which may be applied within healthcare settings, including text summarization. In this study, we evaluate the performance of two LLMs (GPT-4 and GPT-3.5-turbo) in generating encounter summaries for 100 randomly selected Emergency Department visits at the University of California, San Francisco. We found that LLMs could generate accurate encounter summaries, but were liable to hallucination and omission of clinically relevant information. Inaccuracies and hallucinations were most commonly found in the Plan sections of LLM-generated summaries, while clinical omissions were concentrated in text describing patients' Physical Examination findings or History of Presenting Complaint. Individual errors on average had a low potential for harm. Our results provide an understanding of the location and type of errors found in LLM-generated clinical text, helping to facilitate clinician review of such content and prevent patient harm.

## Introduction

Clinical documentation is an essential part of high-quality patient care [1,2]. However, in recent years there has been an increase in the complexity of clinical documentation as a result of the transition from paper-based to electronic health records (EHRs) [3]. This has had downstream effects on the amount of time physicians spend on the EHR, with recent studies suggesting that every hour of direct clinical time spent with patients is associated with 2 extra hours of EHR documentation [4,5]. This concerning increase in EHR burden is a significant contributing factor to the rising prevalence of physician burnout, which may lead to a reduction in the overall quality of patient care [6–9].

A foundational element of clinical documentation is the patient encounter summary, created following both Emergency Department (ED) visits and inpatient hospital admissions. Encounter summaries serve as a critical method of patient information transfer between ED clinicians and outpatient providers [10–12]. However, the process of writing encounter summaries is time-consuming and, consequently, these summaries are often not completed in a timely manner or finished at all [12,13]. This is problematic given that the timeliness and availability of encounter summaries is associated with patients' readmission rates, with the absence of an encounter summary associated with a 79% increased rate of 7-day readmission and 37% increased rate of readmission within 28 days [13]. The AHRQ identifies the lack of adequate post-discharge summarization and communication as primary reasons for ED discharge failures [14].

The recent introduction of large language models (LLMs) such as ChatGPT has led to renewed focus on the use of natural language processing (NLP) to improve both quality and efficiency in healthcare [15]. LLMs possess a range of capabilities which may be applied to the clinical domain, one of which is text summarization.

Chan and Mark Zuckerberg, the Barbara and
Gerson Bakar Foundation, and in the recent
past, the March of Dimes, Juvenile Diabetes
Research Foundation, California Governor's
Office of Planning and Research, California
Institute for Regenerative Medicine, L'Oreal, and
Progenity. None of these entities had any bear-
ing on the design of this study or the writing of
the manuscript. No other authors have conflicts
of interest to disclose.

Previous reports have evaluated the potential use of LLMs in summarizing scientific literature, radiology reports, patient problem lists and doctor-patient conversations, with varying success [16,17]. However, there has been limited research on the ability of LLMs to summarize information from a patient's hospital encounter into an encounter summary [18]. As ambient AI scribes and other LLM-based tools begin to be used for summarization of patient encounters within healthcare settings [19], rigorous evaluations of the accuracy of these technologies are urgently needed.

In this study, we investigate the performance of two state-of-the-art LLMs, GPT-4 and GPT-3.5-turbo, in generating ED encounter summaries and evaluate the prevalence and type of errors across each section of the summary.

## Methods

The UCSF Information Commons contains deidentified structured clinical data as well as deidentified clinical text notes, with externally certified deidentification as previously described [20]. The UCSF Institutional Review Board determined that this use of deidentified data within the UCSF Information Commons is not human participants research and, therefore, was exempt from further approval and informed consent. This study was completed according to a prospectively developed protocol (S1 Protocol).

We identified all adult patients discharged from the University of California, San Francisco (UCSF) ED from 2012 to 2023 with an ED clinician note present within Information Commons (Fig 1). If more than one Emergency Medicine (EM) clinician note was available for a particular ED visit, the earliest note was selected as subsequent notes were often attending attestation notes. In the case of multiple notes with the same chart time, the longest note (by character count) was selected. Clinical notes were minimally preprocessed – only line breaks and extra spaces were removed. Software packages incorporating a series of regular expressions were created and used to examine the structure of notes, confirming the presence/absence of the following note headers: 'Chief Complaint' (274,983/278,629 notes); 'Review of Systems' (263,219/278,629 notes); 'Physical Exam' (276,834/278,629 notes); 'ED Course' (245,900/278,629 notes); and 'Initial Assessment' (139,838/278,629 notes). Notes which did not contain appropriate history, physical examination and assessment/plan sections were excluded. Each note was tokenized using the OpenAI Tiktoken tokenizer [21]. Notes containing ≥3500 tokens were excluded to allow sufficient tokens for the GPT-3.5-turbo API response to be completed within the model's 4096 token context window, which was the shortest context window of the models used. Patients who were admitted to hospital from the ED were identified from the structured electronic health record and excluded so that only patients discharged from the ED were included in our cohort.

Next, we randomly selected two n = 100 samples to be used as the *development* and *test* sets. All prompt engineering and resident annotator training was conducted on the *development* set, while evaluation was conducted on the held-out *test* set. Using the secure, HIPAA-compliant, UCSF Versa Application Programming Interface (API) on Microsoft Azure, we prompted both GPT-3.5-turbo (model = 'gpt-3.5-turbo-0613', role = 'user', temperature = 0; all other settings at default values) and

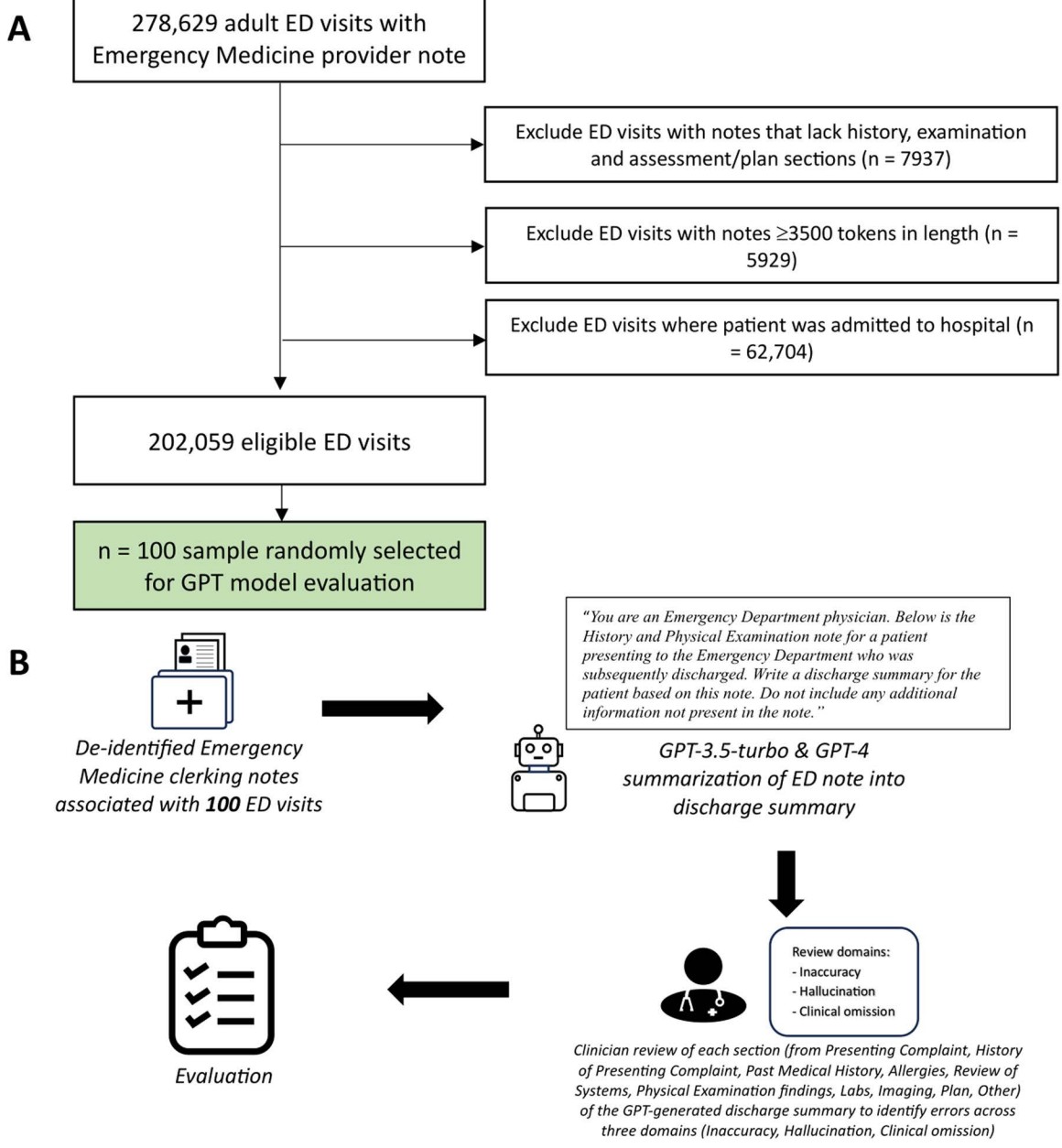

**Fig 1. A) Flowchart of included emergency department (ED) visits.** B) Study workflow.

GPT-4 (model = 'gpt-4-0613', role = 'user', temperature = 0; all other settings at default values) to summarize the full ED clinician note into an encounter summary. Although updated models have since been released, GPT-3.5-turbo and GPT-4 were the latest state-of-the-art models available at the time of study inception. The following prompt was used, followed by the corresponding note for each patient, denoted by triple quotation marks: *"You are an Emergency Department physician. Below is the History and Physical Examination note for a patient presenting to the Emergency Department who was subsequently discharged. Write a discharge summary for the patient based on this note. Do not include any additional information not present in the note. \n\n """ Note text """ ".*

The GPT-3.5-turbo and GPT-4 generated encounter summaries (see S1 Text for examples) were evaluated by two independent EM resident reviewers (from AL, FC, KB, KP, TT) in accordance with the protocol. Initial rates of inter-reviewer agreement were over 90% (S1 Table). Disagreements were resolved by consensus and, if required, by an attending EM physician reviewer (AK). We selected three evaluation criteria for review: 1) Inaccuracy of LLM-summarized information; 2) Hallucination of information; 3) Omission of relevant clinical information. An inaccuracy refers to information that is not factual and/or is contradicted by the original ED clinician note. Hallucination refers to the fabrication of information in the encounter summary that is not present in the original ED clinician note. We distinguish hallucinations from inaccuracies by specifying that, for LLM-generated information to be classified as a hallucination rather than inaccuracy, its correct value must not be present in the original text. Omissions refer to information from the ED clinician note that the reviewer deemed relevant for inclusion in the encounter summary but was not included. The following aspects of a patient's ED visit were evaluated for the presence of inaccuracies, hallucinations, and omissions: Presenting complaint; History of presenting complaint; Past medical history; Allergies/contraindications; Review of systems; Positive examination findings; Laboratory test results; Radiological investigations; Plan; Other notable events during ED stay (if any). On identifying each error, reviewers were additionally asked to provide a brief explanation for their reasoning, which was subsequently manually classified into subgroups of errors within each of the above three evaluation criteria.

To examine the safety of GPT-4-generated encounter summaries, we conducted a post-hoc analysis of the potential for harm of all errors identified in the GPT-4-generated summaries. Each error was assigned a potential harmfulness score by an attending EM physician reviewer (AK), using the Agency for Healthcare Research and Quality (AHRQ) Common Format Harm Scale, adapted to reflect the potential for harm rather than actual harm: 0 – No potential for harm, 1 – Potential for emotional distress or inconvenience (mild and transient anxiety or pain or physical discomfort), 2 – Potential for requiring additional treatment, 3 – Potential for temporary harm (bodily or psychological injury, but likely not permanent), 4 – Potential for permanent harm (lifelong bodily or psychological injury or increased susceptibility to disease), 5 – Potential for lifelong bodily or psychological injury or disfigurement, 6 – Potential for severe permanent harm, 7 – Potential for death [22,23].

### Statistical analysis

For both the GPT-3.5-turbo and GPT-4 encounter summaries, counts of each error (Inaccuracy, Hallucination or Omission) across each section (Presenting complaint; History of presenting complaint; Past medical history; Allergies/contraindications; Review of systems; Positive examination findings; Laboratory test results; Radiological investigations; Plan; Other notable events during ED stay [if any]) relating to the ED visit were collated and reported in a descriptive analysis. The median word count with interquartile range (IQR) for the original EM clinician notes, alongside both the GPT-4-generated and GPT-3.5-turbo generated summaries was calculated. To evaluate encounter summary readability, the average Flesch-Kincaid Reading Ease Score (FRES) and Flesch Kincaid Grade Level (FKGL) was calculated for each GPT model output. Median word counts and FRES/FKGL values were compared using the Mann-Whitney U test against the null hypothesis that there is no significant difference between GPT-4-generated and GPT-3.5-turbo-generated encounter summaries. Categorical variables were compared using the Chi-squared test. $P < 0.05$ was significant. Analyses were performed in Python and R.

### Results

From 202,059 eligible ED visits with an EM clinician note, we randomly sampled 100 for LLM-generated summarization and then expert-driven evaluation (Fig 1; Table 1). The average length of the original EM clinician notes summarized by the GPT models was 802.5 words (IQR 643.5-1053.25) (S1 Fig). GPT-4-generated encounter summaries (median word count = 235 words, IQR 205–264) were shorter than those generated by GPT-3.5-turbo (median word count = 369.5 words, IQR 307.75-445) (S2 Fig; Mann-Whitney U, $p < 0.001$). The average Flesch-Kincaid Grade Level for GPT-4-generated

PLOS Digital Health

**Table 1. Patient demographics in n = 100 sample of Emergency Department encounters randomly selected for GPT-3.5-turbo and GPT-4 encounter summary generation. ED = Emergency department; ESI = Emergency Severity Index; IQR = interquartile range.**

| Variable | Category | Number of patients, n |
|---|---|---|
| *Sex* | Male | 44 |
| | Female | 56 |
| *Race/ethnicity* | White | 39 |
| | Asian | 20 |
| | Black or African American | 18 |
| | Latinx | 11 |
| | Other | 4 |
| | Native Hawaiian or Other Pacific Islander | 3 |
| | Southwest Asian and North African | 3 |
| | Unknown/Declined | 2 |
| *Age, median (IQR)* | 48.1 years (37.4 – 67.9) | |
| *ESI Acuity Level* | Urgent | 54 |
| | Less Urgent | 27 |
| | Emergent | 16 |
| | Non-Urgent | 2 |
| | Unspecified | 1 |
| *Discharge disposition* | Home or Self Care | 95 |
| | Skilled Nursing Facility | 2 |
| | Other | 3 |

summaries was lower (FKGL = 10.0, IQR 9.5-11.1) than for GPT-3.5-turbo-generated summaries (FKGL = 10.7, IQR 9.7-11.7) (Mann-Whitney U, p = 0.02), indicating greater readability of GPT-4-generated encounter summaries. This was also reflected in the Flesch Reading Ease Scores, with GPT-4 summaries (FRES = 48.6, IQR 41.0-52.0) having a higher score on average than GPT-3.5-turbo summaries (FRES = 46.7, IQR 39.7-49.5), though this did not meet statistical significance (Mann-Whitney U, p = 0.10).

Overall, GPT-4-generated encounter summaries contained fewer errors than GPT-3.5-turbo-generated summaries across all three domains (Fig 2). In total, 33% of summaries generated by GPT-4 and 10% of those generated by GPT-3.5-turbo were entirely error-free across all evaluated domains. Summaries generated by GPT-4 were mostly accurate, with inaccuracies found in only 10% of cases. However, 42% of the summaries exhibited hallucinations and 47% omitted clinically relevant information. This compares to 36% of GPT-3.5-turbo summaries containing an inaccuracy, with 64% and 50% of the predecessor model's summaries containing hallucinations and clinical omissions, respectively. Initial inter-reviewer agreement rates were 95.8%, 93.6% and 91.9% for inaccuracy, hallucination and omission errors, respectively, prior to consensus agreement (S1 Table).

Error rate by domain and discharge summary section is shown in Fig 3. The few inaccuracy errors identified in GPT-4-generated encounter summaries predominantly occurred in the *Plan* section of the summary (n = 4). When comparing GPT-3.5-turbo and GPT-4 models, there was a notable improvement in the accuracy of reporting patients' *Past Medical History*, in which 10% of GPT-3.5-turbo summaries contained an error compared to only 1% of GPT-4 summaries. Most hallucination errors, across both GPT-3.5-turbo and GPT-4 models, occurred in either the *Plan* or *Other* sections of the summary, with GPT-4 recording 36% fewer hallucinations in these sections than GPT-3.5-turbo. Omissions were most frequently present in the *Physical Examination* section for both GPT-4 (20%) and GPT-3.5-turbo (18%) summaries, followed by the *History of Presenting Complaint* section (10% of GPT-4 summaries vs 17% of GPT-3.5-turbo summaries).

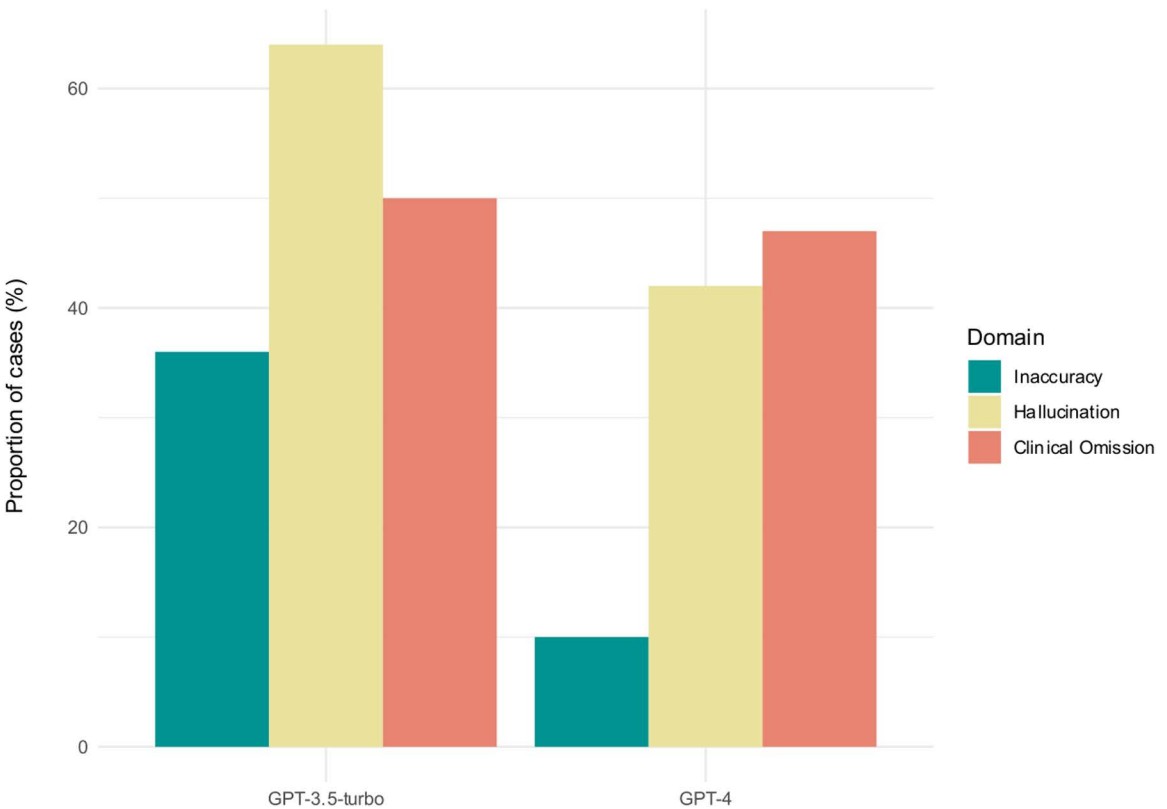

**Fig 2. Proportion of encounter summaries with 1 or more error identified by clinical reviewers in each of the three domains evaluated: 1) Inaccuracy, 2) Hallucination and 3) Clinical Omission.**

In a post-hoc analysis of the potential for harm of errors identified across the GPT-4-generated encounter summaries, the mean potential harmfulness score across all errors was 0.57 (SD 1.11) out of 7 on the adapted AHRQ Common Format Harm Scale. Stratified by error type, inaccuracy errors (mean 0.70, SD 1.25) had the highest potential for harm, followed by omission (0.67, SD 1.08) and hallucination (0.42, SD 1.11) errors. Three errors were scored 4 ('Potential for permanent harm') or greater: one hallucination error was scored at 6 ('Potential for severe permanent harm') - here, the LLM hallucinated a redacted part of the note that the 'patient was diagnosed with vertigo, likely due to a stroke' when their actual diagnosis was a migraine. Meanwhile, two omissions were scored at 4 ('Potential for permanent harm') as a result of omitting, in separate summaries, details on a) the activation of code stroke and b) the measured intra-ocular pressure.

Finally, we manually categorized free-text reviewer comments detailing the subtype of each error (Table 2 and S3 Fig). Among the GPT-4 summaries, inaccuracy errors included inaccurate follow-up details (e.g., reviewer comment: *"[the original note states that the] patient had follow-up with GI for colonoscopy.. already scheduled [whereas the GPT summary states the patient was advised to obtain this]"*), inaccurately reporting the interim plan as the follow up plan (e.g., reviewer comment: *"the final plan is listed [by GPT-4] as 'follow-up labs/psych recommendations', but this was the sign-out plan – the final plan was actually: 'safe for discharge'"*) and inaccurate reporting of physical examination findings (e.g., reviewer comment: *"[the GPT summary] states HINTS exam was positive, but is in fact negative"*). The most commonly identified hallucination error subtype was hallucination of information in the note that had been redacted during the de-identification process (n = 15; e.g., reviewer comment: *"redacted portion [of original*

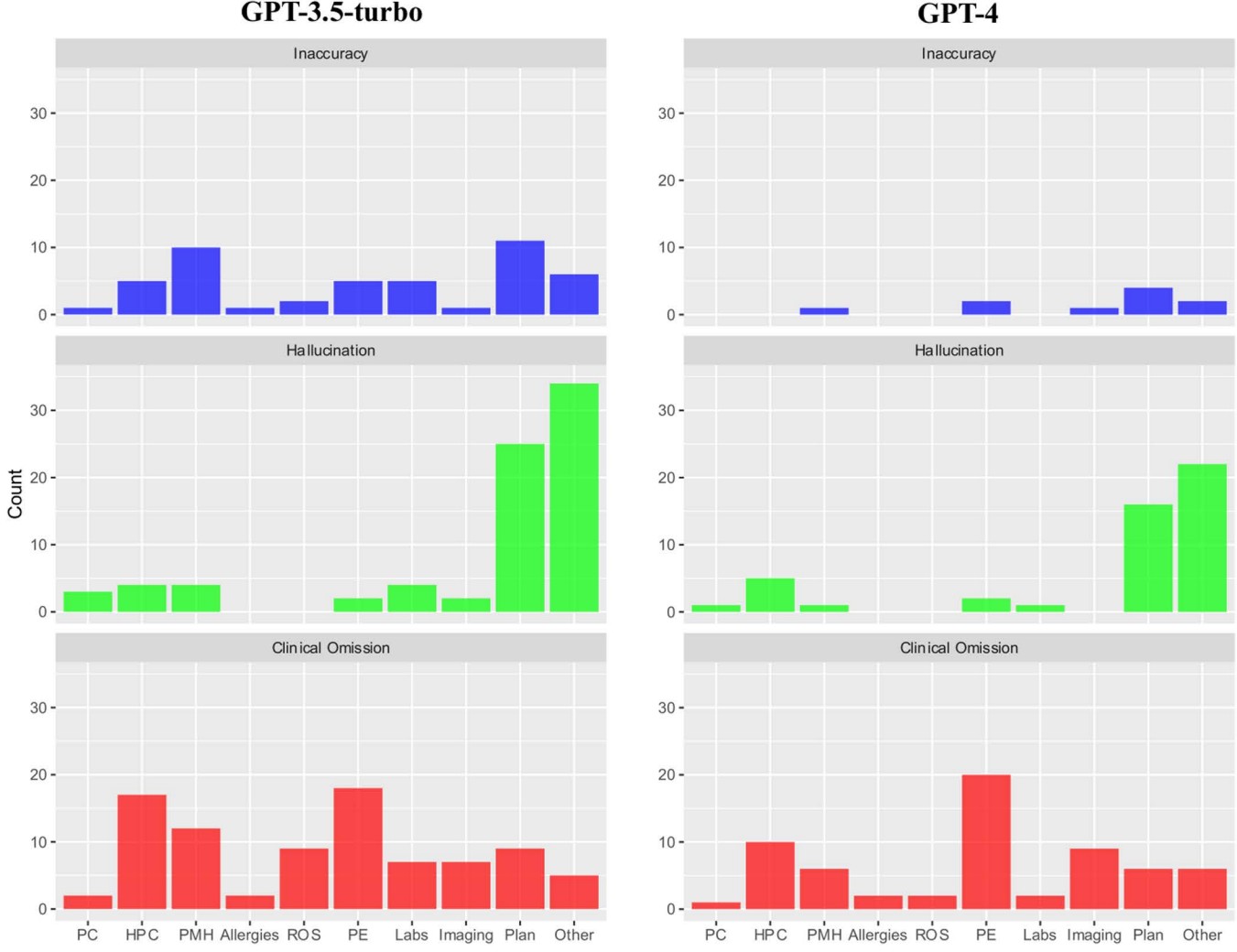

**Fig 3. Breakdown of errors for each domain (Inaccuracy, Hallucination and Clinical Omission) by section of encounter summary.** PC = Presenting Complaint; HPC = History of Presenting Complaint; PMH = Past Medical History; ROS = Review of Systems; PE = Physical Examination.

note] filled in [in GPT summary] as 'headache'"). The next most common hallucinations related to patients' follow up, with GPT-4 either providing details of outpatient specialty follow-up that had not been arranged (n = 11; e.g., reviewer comment: *"[the GPT summary] hallucinated follow-up with Rheumatology and Neurology, though [there is] no mention of this in [the original] note"*), hallucinating ED return precautions (n = 7), and hallucinating follow-up instructions (n = 3; e.g., reviewer comment: *"no instructions to continue current meds or avoid morphine were provided in the original note"*). Meanwhile, examples of the most common omission errors include GPT-4 omitting certain positive physical examination findings (n = 13; e.g., *"[GPT summary] omitted left sided laceration"* or *"[GPT summary] omitted murmur"*), imaging results (n = 8), details of patients' management in ED (n = 7; mostly relating to specialty consults that had taken place) and symptom(s) reported (n = 7; e.g., *"[GPT summary] does not mention Tylenol overdose concern"*). The manually categorized reviewer comments for the GPT-3.5-turbo-generated summaries are shown in S2 Table and S4 Fig.

**Table 2. Manual categorization of overall expert reviewer comments providing further details for each error subtype among GPT-4-generated encounter summaries compared to the ground-truth, original Emergency Medicine provider note. *Comments reported with minor modifications to syntax for improved readability.**

| Error Type | Error category | Example reviewer comment* | Count |
|---|---|---|---|
| Inaccuracy | Inaccurate follow-up details | "[The original note states that the] patient had follow-up with GI for colonoscopy and EGD and hematology follow-up [was] already scheduled [whereas the GPT summary states the patient was advised to obtain this]" | 3 |
| | Inaccurate examination findings | "[The GPT summary] states HINTS exam was positive, but is in fact negative" | 2 |
| | Inaccurately reported the interim plan as the follow-up plan | "The final plan is listed as 'follow-up labs/psych recommendations', but this was the sign-out plan – the final plan was actually: 'safe for discharge'") | 2 |
| | Inaccurately reported patient's management in ED | "Written for but did not get acetaminophen in ED" | 1 |
| | Inaccurately reported imaging as normal | "CT pelvis was not negative" | 1 |
| | Inaccurate social history reported | "States patient is a former smoker, when in fact patient is a former drinker and never smoked" | 1 |
| Hallucination | Hallucinated redacted information | "Redacted portion [of original note] filled in [in GPT summary] as 'headache'" | 15 |
| | Hallucinated outpatient follow-up details | "[The GPT summary] hallucinated follow-up with Rheumatology and Neurology, though [there is] no mention of this in [the original] note" | 11 |
| | Hallucinated ED return precautions | "No return precautions mentioned in non-redacted portion of [original] note" | 7 |
| | Hallucinated medication plan | "[GPT summary] hallucinated plan of continuing medications as prescribed - there was no reference to this in original note" | 3 |
| | Hallucinated primary care physician follow-up details | "[GPT summary] hallucinated PCP follow-up" | 3 |
| | Hallucinated follow-up instructions | "No specific return precautions (fever, chest pain, SOB) provided in [original] note; no instructions to continue current meds or avoid morphine were provided in the original note" | 3 |
| | Hallucinated patient's management in ED | "Patient did not receive Nitro spray in ED" | 3 |
| | Hallucinated cause of symptoms | "Under diagnosis, [GPT summary] states headache is due to post-surgical changes, which was not documented in the initial note" | 1 |
| | Hallucinated patient's diagnosis | "No mention of what final diagnosis was on original note, yet GPT wrote 'likely due to cyst or surgery'" | 1 |
| | Hallucinated symptoms | "[GPT summary] hallucinated patient as having carotid tenderness" | 1 |

*(Continued)*

**Table 2.** (Continued)

| Error Type | Error category | Example reviewer comment* | Count |
|---|---|---|---|
| Clinical Omission | Omission of positive physical examination findings | "Omitted left sided laceration"; "Did not mention contracture"; "Omitted bilateral conjunctival injection"; "Omitted murmur"; "Omitted patient's somnolence and gait stability" | 13 |
| | Omission of imaging performed | "Omitted chest x-ray"; "Omitted MRI results"; "Omission of all radiology results" | 8 |
| | Omission of symptom reported | "Does not mention Tylenol overdose concern"; "No mention of watery diarrhea"; "Omitted that bleeding was seen by ED nurse and pressure dressing was applied" | 7 |
| | Omission of details of patient's management in ED | "Omitted orthopedics consult"; "Omitted gynecology consult"; "Omitted reassessment and repeat check of ambulatory saturation" | 7 |
| | Omission of pertinent negative physical examination findings | "Omitted patient was afebrile"; "Should include that patient had a benign GU exam"; "Should have included benign abdominal exam" | 5 |
| | Omission of details of patient's medication history | "Omitted that she was on antibiotics"; "Omitted estrogen use" | 4 |
| | Omission of details of patient's Past Medical History | "Omitted history of known pulmonary embolus"; "Did not mention clarification on baseline bradycardia (this is significant abnormality that provider contacted PMD to clarify)"; "Omitted key medical history including patient was unable to walk secondary to dizziness" | 4 |
| | Omission of details of patient's allergies | "No mention of allergies" | 2 |
| | Omission of details of patient's Past Surgical History | "Omitted cholecystectomy surgery"; "Omits history of PEG" | 2 |
| | Omission of ECG performed | "Omission of any mention of an electrocardiogram for patient's tachycardia" | 1 |
| | Omission of laboratory tests performed | "Omitted mention of stool studies collected" | 1 |
| | Omission of symptom time course | "Omitted timeline of symptoms – improvement, but woke her up second night in a row" | 1 |
| | Omission of symptom character | "Does not mention that the reason the patient's chest pain is different now is that it is now constant" | 1 |
| | Omission of suspicious injury report | "Omission of suspicious injury report" | 1 |
| | Omission of pertinent normal laboratory test results | "Omitted negative troponins" | 1 |
| | Omission of follow-up information | "Discussion of possible bowel regiment not included" | 1 |
| | Omission that patient declined physical examination | "Refusal of rectal exam" | 1 |
| | Omission of diagnosis | "Did not include the presumptive diagnosis selection (menstrual cramps) amongst the various differential diagnosis entities" | 1 |
| | Omission of code stroke activation | "Omits activation of code stroke" | 1 |
| | Omission of bedside imaging done | "Omits bedside ultrasound (but mentions x-ray)" | 1 |
| | Omission of urinalysis results | "Omitted positive urine drug screen for cocaine (not extremely relevant)" | 1 |

## Discussion

In this cross-sectional study of 100 ED encounters, we found that LLMs could generate accurate encounter summaries but were liable to hallucination and omission of clinically relevant information. Overall, GPT-4-generated summaries contained fewer errors than GPT-3.5-turbo summaries across all three domains, with 10%, 42% and 47% of summaries containing inaccuracies, hallucinations and omissions, respectively. Importantly, errors identified in GPT-4-generated summaries had low potential for harm, with an average score of 0.57 out of 7 on an adapted AHRQ Common Format Harm Scale. GPT-4-generated summaries were also shorter and more readable than those generated by GPT-3.5-turbo, with an average Flesch-Kincaid Grade Level of 10.

The improved performance of GPT-4 compared to GPT-3.5-turbo aligns with prior literature which has shown superior GPT-4 performance across both medical and non-medical tasks [24–26]. Moreover, the fact that GPT-4 summaries contained a lower number of omissions than GPT-3.5-turbo, whilst summarizing the same information in fewer words, suggests increased summary concision that may be welcomed by primary care physicians and others on the receiving end of the transition of care [27].

Although only 33% of summaries generated by GPT-4 were entirely error-free across all domains, a more detailed review of the subtypes of error demonstrated that a majority of hallucinations either related to information redacted in the original note as part of our institution's de-identification process or resulted from GPT-4 hallucinating follow-up instructions and/or return precautions. In the latter instance, such follow-up instructions were often appropriate for the patient's care (as if they were derived from a standard set of precautions associated with the patient's final diagnosis), but because they had not been explicitly mentioned in the original EM provider's note, they were classified as hallucinations in accordance with our pre-specified protocol. After excluding these specific types of errors post-hoc, the proportion of GPT-4 generated summaries considered error-free across all domains increased by 14%, reaching 47% error-free across the three domains.

Meanwhile, there were notable differences in initial inter-reviewer agreement between error type prior to consensus agreement, with 91.9% agreement on the presence of clinical omissions compared to 95.8% and 93.6% agreement for inaccuracies and hallucinations, respectively. This reflects the subjective nature of classifying clinical omissions, where the inclusion of different clinical details may depend on the preference of the discharging clinician. It is possible that, with either dedicated prompt engineering or the addition of few-shot examples during future prompting, clinician-specific preferences of what information ought to be included in each encounter summary may be incorporated to address this.

There is a paucity of existing literature examining the performance of LLMs when generating encounter summaries, either in the Emergency Department or inpatient hospital setting. This is concerning given reports of the recent deployment of ambient artificial intelligence (AI) scribes at a large healthcare organization [19]. In that study, 35 example patient transcripts and encounter summaries generated by the AI scribe were rated using a modified version of the Physician Documentation Quality Instrument, with an average score of 48/50 achieved [19,28]. However, a quantitative analysis of the number and type of errors present was not reported. Meanwhile, a separate study of neurology inpatient encounters showed that Bidirectional Encoder Representations from Transformers (BERT) and Bidirectional and Auto-Regressive Transformers (BART) models could be used to generate summaries which met the standard of care in 62% of cases, but acknowledged that future work should count the number and type of hallucinations in automated summaries [18].

Since clinicians will ultimately be responsible for auditing and modifying clinical documentation produced by LLMs, gaining a thorough understanding of potential error sources in this documentation is critically important. Without a thorough understanding of where errors may occur, there's a risk that errors made by LLMs could be overlooked, potentially harming patient care [29]. Additionally, the increased workload on clinicians to meticulously audit the encounter summary could lead to worsening burnout, potentially negating the benefits of using this technology. Our findings suggest that the location of errors within a LLM-generated encounter summary may vary based on the type of error: inaccuracies and hallucinations are most commonly found within the *Plan* sections of LLM-generated encounter summaries, while the *Physical Examination* and *History of Presenting Complaint* sections should be checked closely for clinical omissions. Future studies

should evaluate the application of LLMs themselves to identify instances of inaccuracy, hallucination and clinical omission errors within LLM-generated clinical documents when compared to the original source documents, allowing clinicians to audit and amend areas that are subject to discordance.

This study has several limitations. First, in this study only the initial EM clinician note was summarized. While this note typically contains the patient's clinical history, physical examination findings, results of investigations performed and overall plan, other pertinent information that is found in notes from other providers, such as physical or occupational therapist recommendations and specialty consult advice, may not have been included in the encounter summary. Future work should evaluate the performance of LLMs in the more complex task of multi-document summarization before deployment to EDs can be considered. Second, due to the time and labor-intensive process of manual expert review, we included 100 randomly selected ED encounters in our sample, which may limit generalizability across different types of patient demographics and presenting symptoms. Notably, our randomly selected sample predominantly consisted of White, Asian or Black/African American patients, with limited representation of other minority groups. As LLM performance continues to be evaluated across different medical tasks, racial and gender bias assessments of these tools must be performed prior to their integration into clinical care [30]. Future work should consider exploring the use of these models in multi-center studies across diverse patient populations. Third, GPT model performance may improve with further iterations of prompt engineering and/or in-context learning. For instance, in comparing GPT-3.5-turbo to GPT-4, there was an enhancement in summarization capabilities across all domains evaluated, including over ED discharge summary length.

## Conclusion

In this cross-sectional study of 100 ED encounters, we found that LLMs could generate accurate encounter summaries but were liable to hallucination and omission of clinically relevant information. Our results suggest that the location of errors within a LLM-generated encounter summary may vary based on the type of error. Individual errors on average had a low potential for harm. A comprehensive understanding of where errors are most likely to occur in LLM-generated clinical text is critically important to facilitate clinician review and revision of such content and prevent patient harm.

## Supporting information

**S1 Fig. Histogram of original Emergency Medicine provider note length among the n = 100 sample of Emergency Department encounters randomly selected for GPT-3.5-turbo and GPT-4 summarization.**
(DOCX)

**S2 Fig. Histogram of word counts of a) GPT-3.5-turbo and b) GPT-4 generated encounter summaries.**
(DOCX)

**S3 Fig. Manual categorization of reviewer comments providing further details for each error subtype (a) Inaccuracy, b) Hallucination, and c) Clinical omission] among GPT-4-generated encounter summaries compared to the ground-truth, original Emergency Medicine provider note.**
(DOCX)

**S4 Fig. Manual categorization of reviewer comments providing further details for each error subtype (a) Inaccuracy, b) Hallucination, and c) Clinical omission] among GPT-3.5-turbo-generated encounter summaries compared to the ground-truth, original Emergency Medicine provider note.**
(DOCX)

**S1 Table. Initial inter-reviewer agreement rates by error type, prior to consensus agreement.** *$p < 0.001$ (Chi-squared test, $\chi^2 = 26.0$).
(DOCX)

**S2 Table. Manual categorization of expert reviewer comments providing further details for each error subtype among GPT-3.5-turbo-generated encounter summaries compared to the ground-truth, original Emergency Medicine provider note.** *Comments reported with minor modifications to syntax for improved readability.
(DOCX)

**S1 Protocol . Study protocol.**
(DOCX)

**S1 Text.** Example GPT-3.5-turbo and GPT-4 generated encounter summaries and original Emergency Medicine note.
(DOCX)

## Acknowledgments

Dr Aaron E. Kornblith is supported by Eunice Kennedy Shriver National Institute of Child Health and Human Development of the National Institutes of Health under award number K23HD110716. The authors acknowledge the use of the UCSF Information Commons computational research platform, developed and supported by UCSF Bakar Computational Health Sciences Institute. The authors also thank the UCSF AI Tiger Team, Academic Research Services, Research Information Technology, and the Chancellor's Task Force for Generative AI for their software development, analytical and technical support related to the use of Versa API gateway (the UCSF secure implementation of large language models and generative AI via API gateway), Versa chat (the chat user interface), and related data asset and services.

Dr Christopher Y.K. Williams had full access to all the data in the study and takes responsibility for the integrity of the data and the accuracy of the data analysis.

## Author contributions

**Conceptualization:** Christopher YK Williams.

**Data curation:** Christopher YK Williams, Jaskaran Bains, Tianyu Tang, Kishan Patel, Alexa N. Lucas, Fiona Chen.

**Formal analysis:** Christopher YK Williams, Brenda Y. Miao.

**Investigation:** Christopher YK Williams, Jaskaran Bains, Tianyu Tang, Kishan Patel, Alexa N. Lucas, Fiona Chen, Brenda Y. Miao, Aaron E. Kornblith.

**Methodology:** Christopher YK Williams, Jaskaran Bains, Brenda Y. Miao.

**Supervision:** Christopher YK Williams, Atul J. Butte, Aaron E. Kornblith.

**Validation:** Jaskaran Bains, Tianyu Tang, Kishan Patel, Alexa N. Lucas, Fiona Chen.

**Writing – original draft:** Christopher YK Williams.

**Writing – review & editing:** Christopher YK Williams, Jaskaran Bains, Tianyu Tang, Kishan Patel, Alexa N. Lucas, Fiona Chen, Brenda Y. Miao, Atul J. Butte, Aaron E. Kornblith.

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
