## [Decision Letter · Decision Letter 0]

PDIG-D-24-00460Evaluating Large Language Models for Drafting Emergency Department Encounter SummariesPLOS Digital Health Dear Dr. Williams, Thank you for submitting your manuscript to PLOS Digital Health. After careful consideration, we feel that it has merit but does not fully meet PLOS Digital Health's publication criteria as it currently stands. Therefore, we invite you to submit a revised version of the manuscript that addresses the points raised during the review process. Please submit your revised manuscript within 60 days Apr 08 2025 11:59PM. If you will need more time than this to complete your revisions, please reply to this message or contact the journal office at digitalhealth@plos.org. Please include the following items when submitting your revised manuscript:* A rebuttal letter that responds to each point raised by the editor and reviewer(s). You should upload this letter as a separate file labeled 'Response to Reviewers '. This file does not need to include responses to any formatting updates and technical items listed in the 'Journal Requirements' section below.* A marked-up copy of your manuscript that highlights changes made to the original version. You should upload this as a separate file labeled 'Revised Manuscript with Track Changes '.* An unmarked version of your revised paper without tracked changes. You should upload this as a separate file labeled 'Manuscript '. If you would like to make changes to your financial disclosure, competing interests statement, or data availability statement, please make these updates within the submission form at the time of resubmission. Guidelines for resubmitting your figure files are available below the reviewer comments at the end of this letter. We look forward to receiving your revised manuscript. Kind regards, David Fraile Navarro, MD PhDAcademic EditorPLOS Digital Health Leo Anthony CeliEditor-in-ChiefPLOS Digital Healthorcid.org/0000-0001-6712-6626 **Journal Requirements:**

2. Please provide an Author Summary. This should appear in your manuscript between the Abstract (if applicable) and the Introduction, and should be 150–200 words long. The aim should be to make your findings accessible to a wide audience that includes both scientists and non-scientists. Sample summaries can be found on our website under Submission Guidelines:

https://journals.plos.org/globalpublichealth/s/submission-guidelines#loc-parts-of-a-submission.

 **Additional Editor Comments (if provided):** Dear Christopher:

Please answer carefully all the comments below provided by reviewers and revise thoroughly your manuscript. Do not hesitate to contact us if you have any questions.**Reviewers' Comments:** Reviewer's Responses to Questions

**Comments to the Author**

1. Does this manuscript meet PLOS Digital Health’s publication criteria ? Is the manuscript technically sound, and do the data support the conclusions? The manuscript must describe methodologically and ethically rigorous research with conclusions that are appropriately drawn based on the data presented.

Reviewer #1: Yes

Reviewer #2: Yes

2. Has the statistical analysis been performed appropriately and rigorously?

Reviewer #1: Yes

Reviewer #2: Yes

3. Have the authors made all data underlying the findings in their manuscript fully available (please refer to the Data Availability Statement at the start of the manuscript PDF file)?

Reviewer #1: Yes

Reviewer #2: No

4. Is the manuscript presented in an intelligible fashion and written in standard English?

Reviewer #1: Yes

Reviewer #2: Yes

5. Review Comments to the Author

Reviewer #1: While this may have been practical for manual review, it limits statistical power and generalizability of findings. More robust approach would involve larger sample size with stratified random sampling to ensure adequate representation across different types of ED encounters, conditions, and complexity levels.

Demographic representation in study presents another important limitation. As acknowledged by authors, random sample predominantly consisted of White, Asian, or Black/African American patients, with limited representation of other minority groups. To address this, researchers should implement purposive sampling strategies to ensure adequate representation across all demographic groups.

Single-institution focus of study, examining notes only from UCSF ED, significantly limits its generalizability. Different institutions have varying documentation styles, EHR systems, and patient populations. Multi-center study including diverse healthcare settings would provide more comprehensive insights into LLM performance across different clinical environments and documentation practices.

Study's reviewer pool was limited to emergency medicine residents with one attending physician for arbitration. This narrow perspective might miss important aspects that different healthcare providers would value in discharge summaries.

While study reports basic agreement rates, it lacks formal statistical analysis of inter-rater reliability. Including kappa statistics or other formal measures would provide more rigorous evidence of consistency and reliability of error identification process. Additionally, categorization of errors, particularly "omissions," appears somewhat subjective. Developing more standardized error classification framework with explicit criteria would improve reproducibility of findings.

Prompt engineering aspect of study appears limited, using single basic prompt without optimization. More comprehensive approach would involve systematic testing of different prompt variations and include few-shot learning examples to optimize LLM performance. This could significantly improve quality of generated summaries and provide valuable insights into best practices for implementing these systems in clinical settings.

While it identifies various types of errors, it doesn't evaluate their potential impact on patient care or quantify efficiency gains versus time needed for human verification. Including severity ratings for errors and analyzing practical implications of implementing LLM-based documentation tools would provide crucial information for healthcare organizations considering adoption of these technologies.

Including multiple LLM versions and vendors, and evaluating performance changes across model iterations would provide more comprehensive understanding of technology's capabilities and limitations. This would also help healthcare organizations make more informed decisions about which systems might best suit their needs.

Starting with pilot study incorporating some of these improvements before conducting larger multi-center study would allow for refinement of methodology and identification of potential implementation challenges.

Reviewer #2: Review of “Evaluating Large Language Models for Drafting Emergency Department Encounter Summaries”.

This article addresses a highly relevant topic given the increasing use of large language models in healthcare. Electronic health record documentation is a significant burden on medical professionals, and leveraging LLMs to streamline this process is promising. However, concerns remain regarding their appropriate deployment and potential risks.

1) The article specifically evaluates GPT-3.5 and GPT-4. It would be helpful to clarify the rationale for selecting these models over other commercially and open source available LLMs. Were they chosen based on prior benchmarking, accessibility, or specific performance attributes relevant to EHR summarization?

2) The study sets the model temperature to 0 to encourage more deterministic outputs. However, GPT models do not default to this setting. Could the authors elaborate on why this specific temperature was chosen and whether alternative values were considered? Exploring different temperatures might provide insights into potential trade-offs between consistency and adaptability in the generated summaries.

3) It would strengthen the results section to include an analysis of whether identified errors had the potential to lead to adverse or dangerous clinical outcomes. Given that errors in EHR summarization can pose clinical risks, it would be beneficial to extend the discussion about potential solutions or mitigation strategies.

4) The post hoc analysis might be better suited within the results section rather than in the discussion.

5) Figure 2: The vertical line in the middle of the bars for different models could be misleading. It might imply an artificial boundary, affecting interpretation. Instead of grouping by model, an alternative visualization could categorize errors by type, allowing for clearer insights into which errors are most prevalent across models.

6) Figure 3: Similar to Figure 2, the visualization could be adjusted for better readability. The combination of horizontal and vertical lines, along with count values, makes comparisons difficult. The GPT-4 white box label overlaps with the first plot, which should be adjusted for clarity.

7) Figure 2 and 3: The graphs are not showing the proportion of cases with more than one error and may mislead to wrong assumptions.

6. PLOS authors have the option to publish the peer review history of their article (what does this mean? ). If published, this will include your full peer review and any attached files.

**Do you want your identity to be public for this peer review?** For information about this choice, including consent withdrawal, please see our Privacy Policy .

Reviewer #1: No

Reviewer #2: **Yes: ** Luis Filipe Nakayama

---

## [Decision Letter · Decision Letter 1]

Evaluating Large Language Models for Drafting Emergency Department Encounter Summaries

PDIG-D-24-00460R1

Dear Dr Williams,

We are pleased to inform you that your manuscript 'Evaluating Large Language Models for Drafting Emergency Department Encounter Summaries' has been provisionally accepted for publication in PLOS Digital Health.

Best regards,

David Fraile Navarro, MD PhD

Academic Editor

PLOS Digital Health

**Additional Editor Comments (if provided):**

**Reviewer Comments (if any, and for reference):**

Reviewer's Responses to Questions

**Comments to the Author**

1. If the authors have adequately addressed your comments raised in a previous round of review and you feel that this manuscript is now acceptable for publication, you may indicate that here to bypass the “Comments to the Author” section, enter your conflict of interest statement in the “Confidential to Editor” section, and submit your "Accept" recommendation.

Reviewer #1: All comments have been addressed

Reviewer #2: All comments have been addressed

2. Does this manuscript meet PLOS Digital Health’s publication criteria ? Is the manuscript technically sound, and do the data support the conclusions? The manuscript must describe methodologically and ethically rigorous research with conclusions that are appropriately drawn based on the data presented.

Reviewer #1: Yes

Reviewer #2: Yes

3. Has the statistical analysis been performed appropriately and rigorously?

Reviewer #1: Yes

Reviewer #2: Yes

4. Have the authors made all data underlying the findings in their manuscript fully available (please refer to the Data Availability Statement at the start of the manuscript PDF file)?

Reviewer #1: Yes

Reviewer #2: Yes

5. Is the manuscript presented in an intelligible fashion and written in standard English?

Reviewer #1: Yes

Reviewer #2: Yes

6. Review Comments to the Author

Reviewer #1: The revisions are comprehensive, and the manuscript is now clearly written and scientifically sound. I have no further comments and recommend acceptance for publication in its current form.

Reviewer #2: The authors have addressed the main concerns, particularly those related to potential risks, with satisfactory clarifications.

I would still recommend enhancing the readability of Figure 3. For instance, in columns with a small number of errors, it is difficult to visually discern whether there is one error or multiple. Additionally, maintaining a consistent color scheme for the error representations in Figures 2 and 3 would improve visual coherence.

7. PLOS authors have the option to publish the peer review history of their article (what does this mean? ). If published, this will include your full peer review and any attached files.

**Do you want your identity to be public for this peer review?** For information about this choice, including consent withdrawal, please see our Privacy Policy .

Reviewer #1: None

Reviewer #2: **Yes: ** Luis Nakayama
